# Spatial Vegetation Patch Patterns and Their Relation to Environmental Factors in the Alpine Grasslands of the Qilian Mountains

Theophilus Atio Abalori [1], Wenxia Cao [1,*], Conrad Atogi-Akwoa Weobong [2], Wen Li [3], Shilin Wang [1] and Xiuxia Deng [1]

1   Grassland Ecosystem Key Laboratory of Ministry of Education, Sino-US Research Centre for Sustainable Grassland and Livestock Management, Grassland Science College of Gansu Agricultural University, Lanzhou 730070, China; abalorit@yahoo.com (T.A.A.); mroseirichard@yahoo.com (S.W.); nanasei2000@gmail.com (X.D.)
2   Faculty of Natural Resources and Environment, University for Development Studies, Tamale P.O. Box 1882, Ghana; conradweobong@yahoo.com
3   Key Laboratory of Development of Forage Germplasm in the Qinghai-Tibetan Plateau of Qinghai Province, Qinghai Academy of Animal Science and Veterinary Medicine of Qinghai University, Xining 810003, China; fosuaama994@yahoo.com
*   Correspondence: caowenxia@foxmail.com; Tel.: +86-130-931-6162

**Abstract:** Globally, grasslands are affected by climate change and unsustainable management practices which usually leads to transitions from stable, degraded and then to desertification. Spatial vegetation patch configurations are regarded as key indicators of such transitions. Understanding the relationships between this grass-land vegetation and its environment is key to vegetation restoration projects. Spatial vegetation patch patterns were chosen across different soil and topographic conditions. Patch numbers, perimeter, and cover of each patch were measured along transects of each patch type. Using field surveys and multivariate statistical analysis, we investigated the differences in vegetation biomass and distribution and soil properties of four typical alpine plant species patches along with a range of environmental and topographic conditions. It was found that topographic conditions and soil properties, particularly soil moisture explained most of the variation in spatial patch vegetation characteristics and thus control vegetation restoration in the alpine grassland. The *Kobresia humilis*, *Blysmus sinocompressus* and *Iris lactea* patches under the drylands recorded small patch sizes, large patch numbers, low connectivity, and large total perimeter per unit area. Generally, species within the high moisture sites recorded small patch numbers, a large fraction of vegetation cover and a small total perimeter per $m^2$. Patches in limited soil moisture areas recorded patch configurations indicating they are unstable and undergoing degradation and therefore need urgent restoration attention to forestall their further degradation and its resultant effect of desertification. These results would provide quantitative easy-to-use indicators for vegetation degradation and help in vegetation restoration projects.

**Keywords:** environmental factors; grassland degradation; patch configurations; spatial; vegetation patch

## 1. Introduction

Alpine grasslands are exceedingly delicate ecosystems that are highly susceptible to global climate change [1] as a result of weather conditions such as low temperature, little rainfall, and low concentrations of oxygen at high elevations [2]. In addition, the Qinghai Tibetan Plateau (QTP) area has been battling substantial climate warming for over five decades which has led to the depletion of the grassland vegetation [3]. Moreover, the alpine grasslands of the QTP have been subjected to anthropogenic activities that led to their serious degradation [4].

Environmental factors such as soil parent material, topography, climate, vegetation, and anthropogenic activities have a major impact on the spatial variability of soil properties in an ecosystem [5]. Topographic-induced microclimate differences can lead to major variations in vegetation characteristics and soil properties which could in turn have drastic effects on the soil structure and functions of the ecosystem [6,7]. To enhance ecological restoration efficiency, it is vital to have a full understanding of the relationships between the typical plant species and the native environment and to recognize major actors that impact their growth, adaptation, and distribution in the fragile alpine grassland ecosystem [8]. Temperature tends to decrease with increasing elevation while moisture increases with increasing elevation because of high precipitation on high altitude.

Several studies on variations in vegetation cover [9], mechanisms and levels of plant adaptation, soil properties, and restoration methods and their impacts have been undertaken [10]. However, most studies concentrated on one component, such as soil quality, slope aspect, or elevation, examining its consequence on vegetation to a large extent, and few examined more factors and looked at the qualitative effect [11]. There has not been any integrated studies on the effects of environmental factors on these typical plants of the alpine meadow grassland ecosystems.

In addition, spatial vegetation patch patterns are indicators of ecosystem stability and health, particularly in the context of climate change and monumental human modifications [12,13]. The stability of alpine grassland ecosystems has a significant role in global carbon cycles and the maintenance of biodiversity [14,15]. Nonetheless, it is delicate and susceptible to external influences, and patchiness is common in the alpine grassland [16,17]. Patchiness is regarded as a reflection of the state and functioning ability of ecosystems [12]. Patch sizes and types indicate major variations in soil attributes, plant biomass, and soil moisture [18,19]. Unequal distribution of water and nutrients and the impact of both livestock and wild animals are the major factors influencing the formation of vegetation patches [20]. There are proven records that patch features such as number, size, area, and connectivity could indicate the degree of stress from external disturbances and signify the degradation stage of grasslands [21,22]. Larger inter-patch distances decrease the ability to hold back propagules for regeneration, hence maximizing erosion hazard [23,24]. Stable patches tend to have large size, small patch numbers, high connectivity, and small total perimeter per unit area. Detecting signs of regime changes is important to foresee and take measures to prevent the desertification of grasslands by developing recovery processes that enhance their sustainability [25,26]. Thus, getting an insight into the features and determining factors of patchiness is a vital step in unveiling the processes and mechanisms of grassland deterioration [16]. However, scientific information concerning the spatial distribution and the factors that drive vegetation patch patterns in the alpine grassland is nonetheless rare.

In this study, field data were obtained regarding changes in the spatial vegetation patch patterns of typical alpine grassland patches across slope aspects and climatic and soil characteristics via field surveys to assess the differences in their attributes and configurations. This study aimed to measure the patch numbers, cover, and perimeter per unit area of four typical alpine species patches (*Kobresia humilis*, *Elymus nutans*, *Blysmus sinocompressus*, and *Iris lactea*) to discover the relationship between patch attributes and selected environmental variables. We hypothesized that vegetation patch attributes would vary across the environmental gradients due to variations in soil moisture. Detrended correspondence analysis (DCA) was also used to estimate spatial vegetation patch distribution. The results would provide scientific information that is helpful for grassland restoration projects in the alpine meadow grassland of the QTP. The study was guided by the following research questions:

1. How soil moisture affects spatial vegetation patch patterns;
2. How topographic conditions affect spatial vegetation patch patterns.

By answering these questions, we aim to gain a better understanding of the effects of soil moisture and topographic conditions on spatial vegetation patch patterns.

## 2. Materials and Methods

### 2.1. Site Description

In this study, field experiments were conducted in the alpine grasslands of the Qilian Mountains in Zhuaxixiulong Township in the Tianzhu Tibetan Autonomous County of Gansu Province of China. The area has a typical alpine climate and is usually cold and wet for most parts of the year. It also has weather conditions such as thin air with low oxygen concentrations, high solar and high ultraviolet radiation. The area has an average annual temperature of 0.13 °C, and average annual rainfall of 414.98 mm, which usually falls from July to September. The growing season is about 120 days, ranging from May to September. The soil is typical alpine chernozem. Typical alpine plant species include *Kobresia humilis*, *Elymus nutans*, *Koeleria pers*, and *Blysmus sinocompressus*. Major grass species within the selected patches with their important values are presented in Table 1. Important values of vegetation species were calculated as follows; Important value (IV) = (Relative coverage + relative height + relative density + relative weight)/4.

**Table 1.** Major grass species and their important values within the selected patches.

| Patch | Species | Family | Important Value |
|---|---|---|---|
| BS | *Blysmus sinocompressus* | Cyperaceae | 1 |
| | *Pedicularis kansuenis* | Scrophulariaceae | 0.03 |
| | *Potentilla anserina* | Rosaceae | 0.13 |
| | *Puccinia chinensis* | Pucciniaceae | 0.19 |
| | *Rheum pumilum* | Polygonaceae | 0.06 |
| ES | *Elymus nutans* | Poaceae | 1 |
| | *Medicago ruthenica* | Fabaceae | 0.25 |
| | *Artemisia sphaerocephala* | Asteraceae | 0.23 |
| | *Oxytropis ochrocephala* | Fabaceae | 0.11 |
| | *Polygala tenuifolia* | Polygalaceae | 0.1 |
| | *Silene gallica* | Caryophyllaceae | 0.14 |
| | *Poa araratica* | Poaceae | 0.28 |
| | *Koeleria pers* | Poaceae | 0.38 |
| | *Kobresia humulis* | Cyperaceae | 0.2 |
| IL | *Iris lactea* | Iridaceae | 1 |
| | *Elymus nutans* | Poaceae | 0.41 |
| | *Polygonum viviparum* | Polygonaceae | 0.23 |
| | *Thalictrum* var. *sibricum* | Ranunculaceae | 0.077 |
| | *Leontopodium nanum* | Asteraceae | 0.15 |
| | *Sphallerocarpus gracilis* | Apiaceae | 0.09 |
| | *Gentianopsis paludosa* | Gentianaceae | 0.14 |
| | *Ranunculus tanguticus* | Ranunculaceae | 0.1 |
| | *Rheum pumilum* | Polygonaceae | 0.14 |
| | *Potentilla anserina* | Rosaceae | 0.11 |
| | *Saussurea japonica* | Asteraceae | 0.058 |
| KH | *Kobresia humilis* | Cyperaceae | 1 |
| | *Medicago ruthenica* | Fabaceae | 0.3 |
| | *Plantago asiatica* | Plantaginaceae | 0.27 |
| | *Elymus nutans* | Poaceae | 0.45 |
| | *Taraxacum mongolicum* | Asteraceae | 0.11 |
| | *Potentilla multifida* | Rosaceae | 0.099 |
| | *Leontopodium nanum* | Asteraceae | 0.098 |
| | *Aconitum carmichaelii* | Ranunculaceae | 0.074 |
| | *Astragalus licentianus* | Fabaceae | 0.09 |
| | *Potentilla discolor* | Rosaceae | 0.027 |

KH = *Kobresia humilis* patch, IL = *Iris lacteal* patch, EN = *Elymus nutans* patch, BS = *Blysmus sinocompressus*.

### 2.2. Experimental Design and Field Sampling

Field surveys were carried out in 4 plant species patches in wetlands (*Kobresia humilis*, *Iris lactea* and *Blysmus sinocompressus*), drylands (*Kobresia humilis*, *Blysmus sinocompressus* and *Iris lactea*), Shady slope (*Elymus nutans*), and sunny slopes (*Elymus nutans*). Representative

samples sites were selected according to the selected grass species patches in wetlands, drylands, and sunny and shady slopes. A total of 10 random sampling quadrats (1 m × 1 m) were established in each site with three replications for a total of 30 quadrats per species patch per site (wet, dry, sunny slope, and shady slope). Figure 1 is a diagram showing how the field sampling was done. Along a transect of 2 m by 30 m in each area, 2 m-by-2 m quadrats were used to record the vegetation patch numbers, perimeter, and percentage cover of vegetation patches. A total of 3 transects were randomly demarcated at each site. Ten adjacent quadrats were laid in each transect. All of the sample plots were evenly distributed from the wet, dry area to the top of the mountains, and included shade slopes and sunny slopes. Landscape pattern analysis was done for patches within the dry and wetlands. GPS was used to obtain the elevation, longitude, and latitude of each sampling site. Quadrats of 1 × 1 m were placed, and plant species, cover, height, and density of each species, as well as above- and belowground biomass were examined. Aboveground biomass was trimmed above the root level by species and collected from each quadrat. Labeled aboveground samples were dried right away at 65 °C until a constant weight was reached. The total C and N contents in the aboveground green biomass of each sample were determined using an elemental combustion analyzer. After collecting the plants, soil samples (10 cm per layer) were collected from 0–30 cm in each quadrat using a soil auger (10 cm internal diameter) to estimate root biomass. Roots were then isolated from soil samples by washing the samples in a 0.5 mm mesh wire. The root samples were dried forthwith in an oven at 65 °C until a constant weight was reached. The bulk density of the soil (10 cm per layer) was measured using a core sampler in each harvested quadrat. In addition, five random soil samples at each depth were collected using a 3.5 cm internal diameter auger, taken from each harvested quadrat and mixed into a single composite sample, and then air-dried for laboratory analysis.

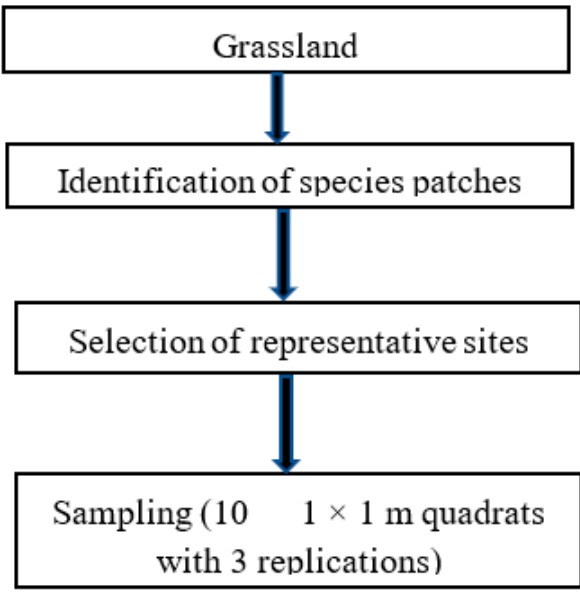

**Figure 1.** Diagram showing how sampling was carried out.

### 2.3. Soil Physical and Chemical Property Analysis

Soil moisture was examined using the oven drying method. Air-dried soil samples were sieved through a 2 mm mesh sieve. The dichromate oxidation method was used to determine soil organic carbon, total N by the Kjeldahl method, total phosphorus by the $HClO_4$–$H_2SO_4$ method, and available phosphorus and available potassium by the molybdenum blue method. All these analyses were carried out as described by Bao [27]. The temperatures at the depths of 0–10 cm, 10–20 cm, and 20–30 cm were measured using a thermocouple probe (LI-8100-203 probe).

Soil pH and EC (soil/water = 1:5) were estimated using a PHS-3C digital pH meter and a DDS-307 conductivity meter, respectively. Soil microbial biomass C and N were determined using the fumigation-extraction method [28]. A 20 g of wet soil (dry weight basis) was fumigated by placing it in a sealed vacuum desiccator containing alcohol-free $CHCl_3$ vapor for 24 hrs. The fumigated base was repeatedly discharged in an aseptic, empty desiccator until the scent of $CHCl_3$ was no longer detectable, and then extracted with 80 mL of 0.5 M $K_2SO_4$ (soil: $K_2SO_4$ = 1:4) for 30 min. The extraction of non-fumigated soil was the same as that of fumigated soil. Soil microbial biomass C and N were calculated as the difference between total organic C and total N in the fumigated and non-fumigated extracts, respectively, with a conversion factor (KEC) of 0.38 and (KEN) of 0.45 [29,30]. $NO_3$-Nand $NH_4$-N: 2 mol KCl in 1 L water, weighed 5 g wet soil, 50 mL KCl was added to the soil and shaken at 180 rpm for 30 min, filtered and pipette 15 mL for analysis.

### 2.4. Statistical Analysis

Vegetation species distribution data were used for biodiversity, and Detrended correspondence analyses (DCA) and Redundancy analysis (RDA) were used to examine relationships between vegetation and environmental parameters. Both were done using the statistical software R.3.6.0 (R Development Core Team, 2016). Multivariate ordination was used to assess the effects of environmental, and soil properties on species composition. The species composition was analyzed and related to 20 variables (soil properties, slope, vegetation cover) using Detrended Correspondence Analysis (DCA). The DCA produced a first axis gradient length of 2.97 SD. We tested its correlation with measured environmental factors using Spearman correlation coefficients. Statistical analyses were also done using the SPSS software program, ver.20.0. Statistically significant differences were set at $p$ values < 0.05. One-way analyses ANOVA were followed by multiple comparisons of least significant differences (LSD test) to compare differences between mean values of vegetation and soil properties within each treatment.

## 3. Results

### 3.1. Patch Distributions

The first two Detrended correspondence analyses (DCA) axes explained 12.9% and 11.9% of the variation in species distribution of the selected grassland patches (Figure 2). The species distribution showed that most of them were clustered in the upper part of the quadrant. *Kobresis humilis* (KH) patch formed species plot 1, and *Blysmus sinocompresuss* patch was species plot 2. Species plots 1 and 2 were vegetation patches found within wetlands. There was then graduation from species plots 1 and 2 to species plot 5 made up of *Kobresia humilis* (Kh) patch within drylands. Species plots 3 and 4 were *Iris lactea* (IL) patches. Whereas species plots 6 and 7 were *Elymus nutans* patches. Species plot 6 was EN within the shady slope whiles that of 7, En within the sunny slope and graduated from the top of the quadrant to the lower-left negative end of the DCA axis. Species scores along the first DCA axis are shown in Table 2. The *Iris lacteal, Blysmus sinocompressus, Elymus nutans*, and *Kobresia humilis* species recorded the highest species scores of above 6. On the other hand, *Pedicularis kansuenis* recorded the least score of −0.8768. Only species with a weight greater than 5 were used.

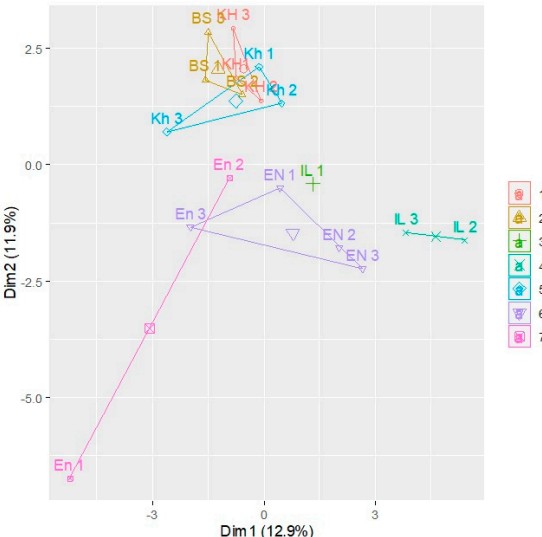

**Figure 2.** Plots of first two axes of DCA of the selected patches. KH, *Kobresia humilis* patch within wetlands; Kh, *Kobresia humilis* patch within drylands; IL, *Iris lactea* patch within drylands; EN, *Elymus nutans* patch within shady slope; En, *Elymus nutans* patch within sunny slope; and BS, *Blysmus sinocompresuss* patch within wetlands.

**Table 2.** Species scores along the first DCA axis. Species with weight greater than 5 were used.

| Species | Score |
|---|---|
| *Pedicularis kansuenis* | −0.8768 |
| *Saussurea japonica* | −0.1569 |
| *Rheum pumilum* | −0.1534 |
| *Thalictrum* var. *sibricum* | 0.2365 |
| *Aconitum carmichaelii* | 0.2856 |
| *Sphallerocarpus gracilis* | 0.7895 |
| *Astragalus licentianus* | 0.8257 |
| *Leontopodium nanum* | 0.9854 |
| *Potentilla multifida* | 1.2980 |
| *Polygala tenuifolia* | 1.4216 |
| *Ranunculus tanguticus* | 1.5465 |
| *Oxytropis ochrocephala* | 1.5980 |
| *Taraxacum mongolicum* | 1.6870 |
| *Potentilla anserina* | 1.8452 |
| *Silene gallica* | 1.9673 |
| *Gentianopsis paludosa* | 2.1093 |
| *Artemisia frigida* | 2.2451 |
| *Puccina chinensis* | 2.3852 |
| *Stipa aliena* | 2.4170 |
| *Poa pova* | 2.4862 |
| *Polygonum viviparum* | 2.6214 |
| *Artemisia sphaerocephala* | 2.7452 |
| *Medicago ruthenica* | 3.0289 |
| *Poa araratica* | 3.4538 |
| *Potentilla discolar* | 3.6832 |
| *Plantago asiatica* | 3.8945 |
| *Aster alpinus* | 4.1268 |
| *Kobresia pygmea* | 4.4320 |
| *Poa annua* | 4.5021 |
| *Potentilla bifurca* | 4.7358 |
| *Iris lactea* | 6.2138 |
| *Blysmus sinocompressus* | 6.4350 |
| *Elymus nutans* | 6.5842 |
| *Kobresia humilis* | 6.6085 |

### 3.2. Numbers, Cover, and Perimeters of Vegetation Patches within Selected Environmental Gradients

Generally, species within the high moisture sites recorded small patch numbers, large fraction of vegetation cover, and small total perimeter per m². The IL patch within the high soil moisture area recorded the highest patch number per m² and total perimeter per m². Species within the low moisture sites had high patch numbers, low fraction of vegetation cover, and large total perimeter per m² indicating low connectivity. Within the low moisture sites, the KH patch had the highest patch number, and the IL patch recorded the highest fraction of vegetation cover and the largest total perimeter per m². The EN patch under the sunny slope had a high patch number and total perimeter per m² while the shady slope recorded a low patch number per unit area and a small total perimeter. Generally, within the same species across the environmental gradient, those in the high soil moisture site recorded small patch numbers, large fraction of vegetation cover, and small total perimeter per unit area indicating they are in stable states (Figure 3). In addition, patches within the high soil moisture sites referred to as wetlands recorded high mean patch sizes compared to the low soil moisture sites referred to as drylands. On the other hand, patches within the drylands recorded larger perimeter- area ratio distribution (PARA_MN) (Table 3).

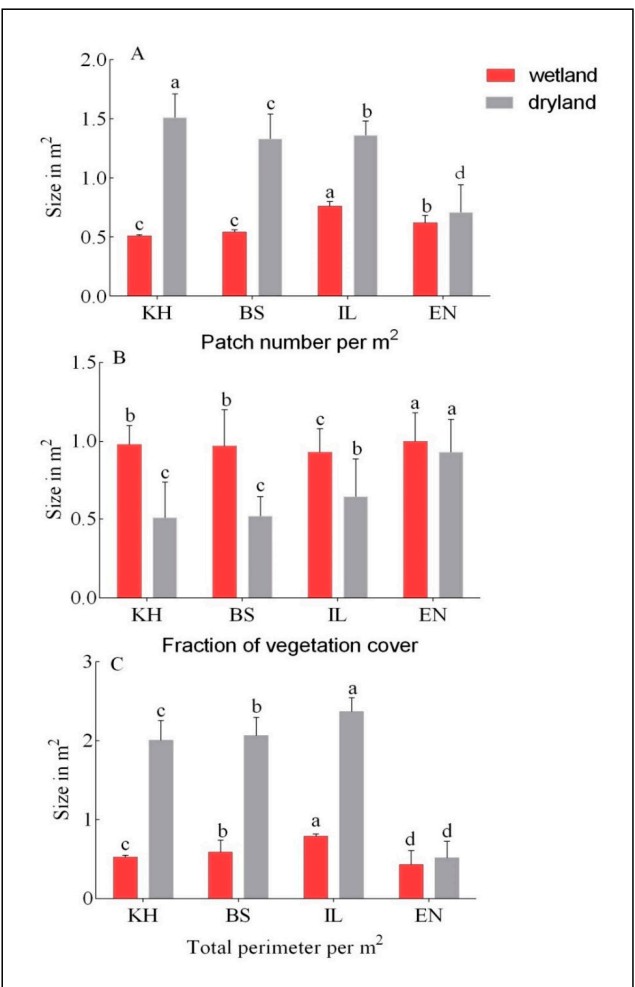

**Figure 3.** Spatial configuration of vegetation patches. Number per m² (**A**), total perimeter per m² (**B**), and cover fraction (**C**) of vegetation patches across different topographic and soil conditions. Means ± SE in different types of vegetation patches is given. Bars sharing the same letters are not different at *p* < 0.001 level. KH; Kobresia patch, BS; *Blysmus sinocompressus*, IL; *Iris lactea*, EN; *Elymus nutans*. Sites with inadequate soil moisture are referred to as drylands and vice versa.

**Table 3.** Landscape pattern indices for the selected sites.

| Site | Index | Min | Max | Mean | Std | Coefficient of Variation (%) |
|---|---|---|---|---|---|---|
| Wetlands | MPS | 0.98 | 2.50 | 1.74 | 0.17 | 8.62 |
| | PARA_MN | 24.68 | 148.09 | 84.29 | 20.01 | 23.74 |
| | SHAPE_MN | 1.34 | 2.94 | 1.67 | 0.1 | 5.98 |
| | PAFRAC | 1.32 | 1.75 | 1.53 | 0.08 | 5.23 |
| Drylands | MPS | 0.26 | 1.27 | 0.76 | 0.13 | 17.11 |
| | PARA_MN | 28.15 | 182.95 | 105.55 | 25.06 | 23.74 |
| | SHAPE_MN | 1.33 | 2.95 | 1.69 | 0.27 | 15.98 |
| | PAFRAC | 1.45 | 3.78 | 2.63 | 0.15 | 5.70 |

MPS—Mean patch size; PARA_MN—Mean perimeter- area ratio distribution; SHAPE_MN—Mean shape index distribution; PAFRAC-Perimeter—area fractional dimension index.

### 3.3. Impact of Slope Aspect on Vegetation Patch Distribution

The plant coverage on the shady slope was significantly higher than that of the sunny slope. Vegetation density, above and below-ground biomass was higher on the shady slope than on the sunny slope. The KH patch under the dryland recorded the highest values of the Shannon–Wiener diversity index and Simpson's index of diversity. The lowest value of Shannon diversity index was recorded at the BS patch under the dryland site while the lowest Simpson's index of diversity was equally recorded at the same site (Table 4).

**Table 4.** Vegetation parameters and biodiversity of selected grassland patches.

| Site | P | Height | Coverage | Density | BGB | AGB | Shannon | Simpson | Evenness | Richness |
|---|---|---|---|---|---|---|---|---|---|---|
| Wet land | KH | 20.40 ± 4.20 [de] | 94.66 ± 1.20 [b] | 775.33 ± 82.10 [b] | 82.00 ± 4.92 [a] | 167.57 ± 25.22 [a] | 2.65 ± 0.22 [ab] | 0.92 ± 0.17 [a] | 0.83 ± 0.05 [a] | 9.11 ± 2.66 [a] |
| Wet land | BS | 14.23 ± 0.31 [f] | 96.16 ± 1.45 [b] | 1495.66 ± 95.71 [a] | 70.29 ± 4.80 [b] | 93.68 ± 2.01 [c] | 1.98 ± 0.22 [c] | 0.85 ± 0.03 [ab] | 0.63 ± 0.02 [b] | 7.33 ± 2.35 [bc] |
| Wetland | IL | 62.41 ± 0.45 [b] | 97.22 ± 0.38 [b] | 99.56 ± 0.74 [d] | 44.63 ± 0.48 [c] | 150.12 ± 0.89 [ab] | 2.71 ± 0.12 [ab] | 0.93 ± 0.04 [a] | 0.84 ± 0.07 [a] | 9.0 ± 2.76 [a] |
| Dryland | BS | 11.21 ± 0.67 [f] | 81.34 ± 0.75 [c] | 98.18 ± 0.59 [d] | 50.67 ± 0.84 [bc] | 79.82 ± 0.84 [d] | 1.81 ± 0.14 [c] | 0.78 ± 0.06 [c] | 0.66 ± 0.08 [ab] | 7.34 ± 2.45 [bc] |
| Dry land | IL | 51.36 ± 0.54 [c] | 95.66 ± 1.85 [b] | 99.00 ± 9.45 [d] | 38.77 ± 0.90 [d] | 133.11 ± 11.30 [b] | 2.60 ± 0.18 [ab] | 0.92 ± 0.13 [a] | 0.85 ± 0.07 [a] | 9.22 ± 1.86 [a] |
| Dry land | KH | 26.38 ± 2.09 [d] | 75.66 ± 2.90 [d] | 56.66 ± 8.87 [e] | 28.41 ± 0.96 [e] | 90.37 ± 24.68 [c] | 3.12 ± 0.01 [a] | 0.95 ± 0.10 [a] | 0.82 ± 0.08 [a] | 9.05 ± 2.65 [a] |
| Shady slope | EN | 76.93 ± 2.06 [a] | 100.00 ± 9.70 [a] | 125.33 ± 11.46 [c] | 24.53 ± 0.42 [e] | 150.81 ± 3.50 [ab] | 2.44 ± 0.12 [ab] | 0.91 ± 0.01 [a] | 0.76 ± 0.06 [ab] | 7.55 ± 1.24 [bc] |
| Sunny slope | EN | 74.35 ± 2.31 [a] | 89.00 ± 6.24 [bc] | 99.00 ± 9.45 [d] | 23.29 ± 0.81 [e] | 133.11 ± 0.30 [ab] | 2.45 ± 0.16 [ab] | 0.91 ± 0.01 [a] | 0.79 ± 0.89 [ab] | 7.68 ± 1.35 [bc] |

Note: Data are presented as the mean ± SD; Different small letters in the same column mean significant difference at 0.05 level. P = patch type; KH = *Kobresia humilis*; BS = *Blysmus sinocompressus*; IL = *Iris lactea*; EN = *Elymus nutans*; BGB = below ground biomass; AGB = above ground biomass; Shannon = Shannon–Wiener diversity index; Simpson = Simpson's index of diversity.

### 3.4. Changes in Vegetation Biomass as a Result of Soil Moisture

The maximum coverage was found in the BS patch which was located in an elevated moisture environment and the lowest was found in the KH patch within the low moisture site. Within the wetland sites, the BS patch had a significantly higher density compared to KH. KH patch however had significant aboveground biomass. The maximum Shannon–Weiner and Simpson's index of diversity were found in the KH patch (Table 4).

Redundancy analysis (RDA) was used to elucidate the connections between the soil properties, topographic features, and vegetation distribution. Vegetation parameters were set as response variables while soil properties and topographic features were set as independent variables for the RDA analysis (Figure 4). Soil TK, SOC, TN, and EC were positively correlated and contributed immensely to the first RDA axis. AGB was positively correlated with TK whilst vegetation coverage was positively correlated with SOC. BGB and MC

were positively correlated and both were positively correlated with density. Porosity and temperature were positively correlated with elevation.

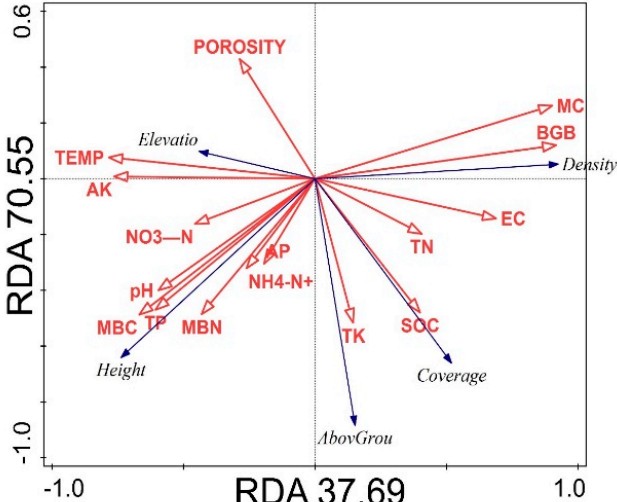

**Figure 4.** RDA for the relationships between vegetation parameters, soil, and topographic features. Note; TN, total nitrogen; TK, total potassium; AP, available phosphorus; AK, available potassium; TP, total phosphorus; MBC, soil microbial biomass carbon; MBN, soil microbial biomass nitrogen; EC, soil electrical conductivity; SOC, soil organic carbon; AbovGrou, above ground biomass; MC, soil moisture content; BGB, below ground biomass; Temp, temperature; Elevatio, elevation.

NO$_3$-N, pH, MBC, TP, MBN, AP, and NH$^+_4$-N were positively correlated and had a positive correlation with plant height. Temperature, AK, porosity, and elevation had a negative correlation with vegetation parameters. Soil moisture explained most of the variations in vegetation properties while BGB explained most of the variations in soil properties (Table 5). MC was significantly positively correlated with BGB and vegetation density.

**Table 5.** Contribution of the various variables to the variations in vegetation and soil properties.

| Name | Explains % | Pseudo-F | *p* |
|---|---|---|---|
| MC | 33.4 | 8 | 0.002 |
| BGB | 32.7 | 7.8 | 0.002 |
| MBC | 30.1 | 6.9 | 0.002 |
| pH | 25 | 5.3 | 0.002 |
| TEMP | 24.7 | 5.3 | 0.004 |
| TP | 24.4 | 5.2 | 0.002 |
| AK | 23.1 | 4.8 | 0.004 |
| NO$_3$-N | 22.8 | 4.7 | 0.004 |
| MBN | 21 | 4.3 | 0.008 |
| EC | 19.3 | 3.8 | 0.006 |
| SOC | 16.6 | 3.2 | 0.008 |
| BD | 11.5 | 2.1 | 0.082 |
| POROSITY | 11.5 | 2.1 | 0.08 |
| NH$_4$-N | 10.3 | 1.8 | 0.126 |
| TK | 10.1 | 1.8 | 0.132 |
| TN | 8.4 | 1.5 | 0.226 |
| AP | 5.5 | 0.9 | 0.438 |

Note: Significance is at *p* = <0.01. MC = soil moisture content, BGB = below ground biomass, MBC = soil microbial biomass carbon, TEMP = soil temperature, TP = total phosphorus, AK = available potassium, NO$_3$-N = Nitrate, MBN = soil microbial biomass nitrogen, EC = soil electrical conductivity, SOC = soil organic carbon, BD = bulk density, NH$_4$-N = ammonium nitrogen, TK = total potassium, TN = total nitrogen, AP = available phosphorus.

### 3.5. Variations in Soil Properties within the Selected Sites

Soil chemical properties for the various sites are shown in Table 6. The highest pH was found in the IL patch and the lowest in the BS patch. The highest mean value of SOC was found in the EN patch on the shady slope whilst the lowest was found in the KH patch under the dryland site. There was generally a reduction in mean values of SOC with respect to increasing soil depth. The maximum mean values of $NH^+_4$-N were found in the KH patch in the wetlands site and the EN patch on the shady slope. TN was significantly higher in the KH patch within the wetland site. Mean values of $NO_3$-N and TP were significantly higher in both the sunny and shady slopes than in the rest of the sites. TK and AP were significantly higher in the KH patch under the wetland than in the rest of the sites. Comparable to the rest of the treatments, significantly higher values of AK, MBN and MBC were recorded in the EN patch under the shady slope. The KH patch within the wetland site recorded the highest mean value of EC.

**Table 6.** Spatial distribution of soil chemical and microbial properties within the selected patches.

| Depth (cm) | Site | P | pH | SOC (g/kg) | NH₄-N (mg/kg) | TN (g/kg) | NO-₃N (mg/kg) | TP (g/kg) | TK (g/kg) | AP (mg/kg) | AK (mg/kg) | MBN (mg/kg) | MBC (mg/kg) | EC(dS/m) |
|---|---|---|---|---|---|---|---|---|---|---|---|---|---|---|
| 0–10 | Wet land | KH | 7.93 ± 0.09 b | 103.87 ± 0.93 c | 35.42 ± 0.11 e | 5.65 ± 0.06 a | 16.53 ± 0.03 a | 0.55 ± 0.01 a | 12.7 ± 0.07 d | 24.43 ± 0.11 e | 127.48 ± 1.09 c | 12.52 ± 0.13 c | 259.79 ± 0.71 a | 211.67 ± 24.45 a |
| | Wet land | BS | 7.39 ± 0.05 a | 95.26 ± 0.20 b | 21.37 ± 0.04 a | 4.77 ± 0.09 b | 21.28 ± 0.06 d | 0.52 ± 0.01 a | 8.03 ± 0.08 b | 15.48 ± 0.11 b | 118.24 ± 0.50 b | 12.42 ± 0.02 c | 312.91 ± 0.41 b | 175.33 ± 35.84 a |
| | Dry land | IL | 8.09 ± 0.00 b | 77.02 ± 0.55 a | 26.67 ± 0.09 b | 4.06 ± 0.03 bc | 16.84 ± 0.03 a | 0.55 ± 0.01 a | 6.94 ± 0.11 a | 14.73 ± 0.08 a | 101.96 ± 1.02 a | 10.89 ± 0.06 a | 348.52 ± 0.96 d | 173 ± 26.35 b |
| | Dry land | KH | 8.00 ± 0.01 b | 76.53 ± 0.77 a | 32.4 ± 0.12 d | 4.54 ± 0.03 cd | 19.12 ± 0.05 b | 0.57 ± 0.01 a | 7.82 ± 0.02 b | 20.34 ± 0.09 c | 243.44 ± 1.81 e | 11.89 ± 0.06 b | 321.06 ± 0.81 c | 120.67 ± 6.98 a |
| | Shady slope | EN | 8.05 ± 0.01 b | 110.56 ± 0.24 e | 35.26 ± 0.11 e | 5.14 ± 0.04 d | 23.67 ± 0.13 d | 0.74 ± 0.03 b | 12.68 ± 0.14 d | 23.57 ± 0.04 d | 252.99 ± 2.69 f | 18.48 ± 0.10 e | 505.69 ± 2.08 e | 154.33 ± 1.86 a |
| | Sunny slope | EN | 8.03 ± 0.02 b | 90.31 ± 0.14 b | 30.48 ± 0.12 c | 4.93 ± 0.02 e | 24.12 ± 0.07 e | 0.78 ± 0.02 b | 10.68 ± 0.07 c | 23.88 ± 0.01 d | 234.46 ± 1.52 d | 17.78 ± 0.09 d | 510.83 ± 1.03 e | 119.67 ± 2.91 a |
| 10–20 | Wet land | KH | 7.94 ± 0.07 b | 83.82 ± 1.39 d | 26.71 ± 0.07 c | 4.68 ± 0.01 b | 15.08 ± 0.06 a | 0.38 ± 0.01 a | 11.49 ± 0.08 d | 17.31 ± 0.18 c | 102.74 ± 1.10 c | 9.83 ± 0.05 c | 204.58 ± 1.67 a | 195.67 ± 12.71 c |
| | Wet land | BS | 7.49 ± 0.03 a | 82.41 ± 0.04 cd | 18.51 ± 0.13 a | 3.97 ± 0.07 ab | 18.28 ± 0.09 c | 0.38 ± 0.00 a | 8.5 ± 0.13 b | 11.3 ± 0.12 b | 90.98 ± 0.17 b | 9.39 ± 0.09 c | 287.2 ± 0.98 c | 99.33 ± 10.47 a |
| | Dry land | IL | 8.1 ± 0.01 b | 60.03 ± 0.09 a | 23.65 ± 0.11 b | 3.81 ± 0.02 a | 15.42 ± 0.06 a | 0.38 ± 0.01 a | 6.95 ± 0.08 a | 9.89 ± 0.04 a | 86.73 ± 1.06 a | 7.69 ± 0.11 a | 256.83 ± 1.98 b | 164.67 ± 16.34 bc |
| | Dry land | KH | 7.99 ± 0.00 b | 68.22 ± 0.24 b | 29.44 ± 0.02 d | 3.90 ± 0.01 ab | 16.63 ± 0.13 b | 0.38 ± 0.00 a | 7.3 ± 0.29 a | 17.26 ± 0.04 c | 190.27 ± 0.17 e | 8.80 ± 0.07 b | 258.97 ± 1.16 b | 146.33 ± 11.05 abc |
| | Shady slope | EN | 8.05 ± 0.02 b | 90.41 ± 0.10 e | 26.60 ± 0.10 c | 4.49 ± 0.06 c | 18.58 ± 0.26 c | 0.45 ± 0.02 b | 12.55 ± 0.03 e | 17.5 ± 0.06 cd | 199.46 ± 0.23 f | 14.56 ± 0.13 e | 417.55 ± 8.81 d | 145.67 ± 4.09 abc |
| | Sunny slope | EN | 8.04 ± 0.01 b | 80.34 ± 0.07 c | 26.50 ± 0.15 c | 4.09 ± 0.02 b | 19.91 ± 0.07 d | 0.49 ± 0.00 b | 10.69 ± 0.05 c | 17.71 ± 0.13 d | 180.95 ± 0.13 d | 13.45 ± 0.09 d | 400.82 ± 1.04 d | 130.33 ± 7.05 ab |
| 20–30 | Wet land | KH | 7.89 ± 0.09 b | 46.57 ± 1.51 b | 25.33 ± 0.14 a | 3.56 ± 0.08 b | 13.07 ± 0.02 b | 0.30 ± 0.00 a | 10.74 ± 0.12 e | 10.12 ± 0.04 c | 95.24 ± 0.73 b | 5.38 ± 0.11 bc | 188.90 ± 0.23 a | 217.33 ± 9.68 d |
| | Wet land | BS | 7.47 ± 0.03 a | 56.54 ± 0.26 d | 16.72 ± 0.15 a | 3.22 ± 0.04 a | 15.33 ± 0.06 e | 0.31 ± 0.01 a | 7.34 ± 0.13 c | 9.39 ± 0.07 e | 74.85 ± 0.41 a | 5.23 ± 0.04 bc | 203.90 ± 1.20 c | 106 ± 8.54 a |

**Table 6.** *Cont.*

| Depth (cm) | Site | P | pH | SOC (g/kg) | NH$_4$-N (mg/kg) | TN (g/kg) | NO-$_3$N (mg/kg) | TP (g/kg) | TK (g/kg) | AP (mg/kg) | AK (mg/kg) | MBN (mg/kg) | MBC (mg/kg) | EC(dS/m) |
|---|---|---|---|---|---|---|---|---|---|---|---|---|---|---|
| | Dry land | IL | 8.06 ± 0.02 [b] | 40.02 ± 0.17 [a] | 15.14 ± 0.07 [a] | 3.17 ± 0.03 [a] | 12.84 ± 0.01 [a] | 0.31 ± 0.00 [a] | 6.13 ± 0.04 [a] | 8.73 ± 0.01 [a] | 74.67 ± 0.95 [a] | 4.86 ± 0.04 [a] | 200.27 ± 0.24 [b] | 198 ± 12.52 [cd] |
| | Dry land | KH | 8.02 ± 0.00 [b] | 45.07 ± 0.30 [b] | 27.61 ± 0.14 [a] | 3.18 ± 0.03 [a] | 13.55 ± 0.02 [c] | 0.31 ± 0.01 [a] | 6.76 ± 0.18 [b] | 10.31 ± 0.10 [c] | 100.26 ± 0.19 [c] | 5.28 ± 0.03 [bc] | 200.74 ± 0.30 [bc] | 159.33 ± 15.49 [bc] |
| | Shady slope | EN | 8.09 ± 0.01 [b] | 53.05 ± 1.39 [cd] | 22.18 ± 0.07 [ab] | 3.45 ± 0.01 [b] | 14.12 ± 0.04 [d] | 0.34 ± 0.02 [ab] | 11.88 ± 0.01 [f] | 11.26 ± 0.05 [d] | 120.08 ± 0.07 [e] | 5.49 ± 0.11 [c] | 298.94 ± 0.97 [d] | 129.66 ± 2.33 [ab] |
| | Sunny slope | EN | 8.06 ± 0.02 [b] | 50.76 ± 0.13 [c] | 24.53 ± 0.21 [ab] | 3.93 ± 0.01 [c] | 15.92 ± 0.01 [f] | 0.37 ± 0.00 [b] | 9.80 ± 0.04 [d] | 11.57 ± 0.10 [d] | 117.08 ± 0.70 [d] | 5.31 ± 0.05 [bc] | 300.43 ± 0.43 [d] | 135.00 ± 2.64 [ab] |

Note: Data are presented as the mean ± SD; Different small letters in the same column mean significant difference at 0.05 level. P = patch type; KH = *Kobresia humilis*; BS; *Blysmus sinnicompressus*, IL; *Iris lacteal*, EN; *Elymus nutants*, SOC; TN = total nitrogen; NO3N; NH4N; TP = total phosphorus; TK = total potassium; AP = available phosphorus; AK—available potassium; MBN = soil microbial biomass nitrogen; MBC = soil microbial carbon; EC = electrical conductivity.

Soil physical properties for the various sites are shown in Table 7. Compared to all the other treatments, the EN patch in the sunny slope recorded a significantly higher mean temperature. Undoubtedly the KH and BS patches under wetland sites recorded elevated soil moisture content (MC) significantly higher than the rest of the sites. However, MC within the KH patch decreased with soil depth but that of BS increased with soil depth. The IL patch recorded a significantly higher BD value and lowest porosity. The KH patch under the dryland site had a significantly higher value of porosity and a significant lower value of BD as compared to the rest.

**Table 7.** Spatial distribution of soil physical properties within the selected patches.

| Depth (cm) | Site | P | TEMP (°C) | MC (%) | BD (g/cm$^3$) | Porosity (%) |
|---|---|---|---|---|---|---|
| | Wet land | KH | 14.33 ± 1.65 [a] | 65.7 ± 1.70 [c] | 1.29 ± 0.06 [b] | 51.19 ± 2.31 [b] |
| | Wet land | BS | 15.9 ± 0.88 [a] | 64.67 ± 0.33 [c] | 1.27 ± 0.04 [b] | 51.95 ± 1.40 [b] |
| 0–10 | Dry land | IL | 16.67 ± 0.82 [ab] | 20.24 ± 1.36 [a] | 1.77 ± 0.03 [c] | 33.33 ± 1.20 [a] |
| | Dry land | KH | 20.23 ± 0.26 [bc] | 29.93 ± 1.67 [b] | 0.97 ± 0.05 [a] | 63.14 ± 1.83 [c] |
| | Shady slope | EN | 18.37 ± 0.41 [ab] | 29.7 ± 1.58 [b] | 1.23 ± 0.06 [b] | 53.71 ± 2.18 [b] |
| | Sunny slope | EN | 24.47 ± 0.75 [c] | 19.54 ± 1.18 [a] | 1.13 ± 0.04 [a] | 57.36 ± 1.52 [bc] |
| | Wet land | KH | 13.3 ± 0.50 [a] | 66.06 ± 0.90 [c] | 1.38 ± 0.12 [ab] | 47.93 ± 4.47 [ab] |
| | Wet land | BS | 13.8 ± 0.40 [a] | 68.67 ± 0.33 [c] | 1.35 ± 0.05 [ab] | 48.91 ± 1.74 [ab] |
| 10–20 | Dry land | IL | 14.53 ± 0.12 [ab] | 19.27 ± 1.03 [a] | 1.49 ± 0.13 [b] | 43.77 ± 4.95 [a] |
| | Dry land | KH | 17.20 ± 0.60 [bc] | 29.55 ± 1.67 [b] | 1.04 ± 0.08 [a] | 60.38 ± 2.72 [b] |
| | Shady slope | EN | 18.53 ± 0.59 [c] | 21.91 ± 1.62 [a] | 1.28 ± 0.04 [ab] | 51.7 ± 1.52 [ab] |
| | Sunny slope | EN | 19.10 ± 0.87 [c] | 19.56 ± 1.65 [a] | 1.25 ± 0.03 [ab] | 52.83 ± 0.99 [ab] |
| | Wet land | KH | 12.56 ± 0.34 [a] | 60.83 ± 2.33 [c] | 1.48 ± 0.14 [b] | 44.15 ± 5.30 [a] |
| | Wet land | BS | 12.93 ± 0.20 [a] | 67.00 ± 0.00 [d] | 1.32 ± 0.06 [ab] | 50.06 ± 2.50 [ab] |
| 20–30 | Dry land | IL | 12.96 ± 0.20 [ab] | 13.65 ± 0.89 [a] | 1.40 ± 0.05 [ab] | 48.30 ± 3.35 [ab] |
| | Dry land | KH | 15.56 ± 0.56 [bc] | 27.19 ± 1.63 [b] | 1.08 ± 0.03 [a] | 59.24 ± 1.43 [b] |
| | Shady slope | EN | 19.93 ± 0.91 [d] | 18.86 ± 0.64 [a] | 1.34 ± 0.06 [ab] | 49.43 ± 2.56 [ab] |
| | Sunny slope | EN | 16.23 ± 0.69 [c] | 15.28 ± 1.22 [a] | 1.23 ± 0.03 [ab] | 56.85 ± 2.13 [ab] |

Note: Data are presented as the mean ± SD; Different small letters in the same column mean significant difference at 0.05 level. P = patch type; KH = *Kobresia humilis*; BS = *Blysmus sinnocompressus*; IL = *Iris lactea*; EN = *Elymus nutans*; TEMP = temperature; MC = moisture content; BD = bulk density.

## 4. Discussion

### 4.1. Effect of Habitat Conditions on Spatial Vegetation Patch Patterns

Alpine grasslands are exceedingly delicate ecosystems that are highly susceptible to global climate change [1]) as a result of weather conditions such as low temperature, low rainfall, and low concentration of oxygen at high elevations [2].

The study showed major differences in soil moisture among the various topographic areas like shady and sunny slopes which are consistent with previous studies [31]. The soil moisture at the sunny slope was lower than on the shady slope which could be attributed to differences in evapotranspiration as a result of differences in solar radiation. Soil moisture was higher at sites near the watercourse. The results also indicate variations in vegetation biomass along elevation gradient and aspect. EN patch tended to have more biomass on the shady slope than on the sunny slope probably due to the differences in soil moisture. Vegetation biomass was significantly affected by topographic features that result in areas of high soil moisture due to their proximity to watercourses than those with low moisture content. The intricate and different connections between soil properties and topographic conditions impact the constitution and biodiversity of grassland vegetation community [32,33] and offer a clear understanding of the features that will make plant species fit or not for local vegetation restoration.

The RDA results indicate that topography and soil properties explain more of the vegetation variation especially soil moisture. In high mountain areas, topographic conditions could explain local vegetation distribution and composition very well [34]. Topographic-induced microclimate differences can lead to major variations in vegetation distribution and soil properties which could in turn have drastic effects on the soil structure and functions of the ecosystem [6,35]. It is therefore not surprising that soil moisture which had the greatest impact on vegetation density was spatially distributed within the study area, interspersed with areas of low and high soil moisture content. Pei et al. [36] found close relationships between elevation, soil moisture, and vegetation. The RDA analysis (Figure 4) showed that vegetation coverage was significantly correlated with SOC. Soil moisture explained 33.4% of the vegetation variation within the study area. Hence understanding the spatial distribution of soil moisture at each site will be central to fruitful grassland restoration of the alpine grassland. Below-ground biomass explained most of the variations in soil properties probably due to its impact on soil bulk density, soil pore space, increasing soil porosity, and hence increase in soil water infiltration rate. Roots have an important function in plant nutrition and water intake, as well as minimizing soil erosion due to rill and gully erosion and increasing soil infiltration ability [37]. They are a significant carbohydrate sink, and their senescence and breakdown release vast quantities of carbon into the soil [38]. Substantial below-ground productivity results in a large supply of organic matter into the soil, which accounts for some of the organic carbon storage in grassland chernozem soils [39]. Below-ground C inputs have a greater impact on SOC formation than above-ground C inputs [40]. Root-derived SOC was around 2.3 times greater than SOC from above-ground crop residues in a long-term agricultural experiment in Sweden [41]. The efficiency of below-ground C inputs into the organic matter pool was 7 to 10-fold higher than that of above-ground C inputs.

The results of the study demonstrate that vegetation patches in areas of high soil moisture had large vegetation patch cover, small patch numbers, and total perimeters per unit area while the reverse holds for areas with low soil moisture. The sunny and shady slopes though did not have elevated soil moisture and also recorded patch attributes closed to the patches located in areas of elevated soil moisture. Apart from grazing being a vital driver of spatial configurations of vegetation patches, less soil water and nutrients also play an important role [20]. Elevated resource limitation leads to spatial reorganization of plants and nutrients in ecosystems with unpleasant environmental conditions and this brings forth the development of localized patterns such as gaps, stripes, and spots [20]. Systematic spatial patterns usually have a small perimeter per unit area and low edge exposure, hence minimizing the hazard of large patches being degraded. Furthermore,

regular patch patterns tend to reduce inter-patch distances, hence promoting connectivity and enhancing favorable feedback between vegetation patches [42]. As soon as resource limitation reaches a threshold the ecosystem switch towards a homogeneous state leading to the complete loss of plants [43]. Abiotic factors, as well as plant-plant interactions, have a role in affecting patch-size distributions, just as they do with regularity. Large patches that facilitate the formation of power-law patch size distributions are encouraged via facilitative interactions [44,45].

Other elements, in addition to plant-plant interactions and aridity, influence vegetation spatial patterns. Even in the absence of more deterministic processes, increases in cover may favor mechanical vegetation clumping due to a shortage of space ([46]. Some abiotic factors, such as rainfall seasonality or soil texture, might exacerbate water stress [47] and hence influence infiltration rates, influencing regional patterns [48,49]. Furthermore, species-specific characteristics may influence spatial pattern formation in a variety of ways. Large species, for example, have a strong influence on the establishment of patch size distributions [50]. Similarly, clonal plants produce predictable patterns [51]. In the IL, KH, and BS patches under declined soil moisture and nutrients had small patch sizes, large patch numbers, and total perimeter per unit area. This implies these patches under the low soil moisture are in an unstable state and undergoing degradation.

Landscape metrics can be used to assess the spatial information of landscape pattern composition, arrangement, and land use/land cover changes [52,53]. Various landscape metrics have been backed by many fields for decades and are extensively employed by researchers and policy-makers in evaluating, monitoring, and predicting landscape trends and land-use changes [12,54]. Because of the more fragmentation of patches within the dryland sites, it is seen that they recorded patch configurations indicating they are unstable and undergoing degradation. This is equally seen in landscape pattern analysis where the mean patch size in the wetlands is larger than those in the drylands. The patches within drylands had a bigger perimeter-area ratio distribution indicating they are unstable and hence have high perimeter and small patch size. The *Iris lacteal*, *Blysmus sinocompressus*, *Elymus nutans*, and *Kobresia humilis* species recorded the highest species scores from the DCA, and this is so because they were the dominant species and the research objects. Therefore, urgent restoration attention must be given to the dry sites which recorded spatial vegetation patch configurations indicating they are unstable and undergoing degradation to prevent them from further degradation and its resultant effect of desertification.

### 4.2. Impact of Soil Properties on Vegetation Patches

Insufficient soil nutrients and moisture bring about stress on vegetation and hence lead to its degradation. Besides grazing as the major biotic driver, the chief abiotic drivers underlying the spatial configurations of vegetation patches are regarded to be limited resources, such as soil water and nutrients [43]. Increased resource scarcity results in spatial reorganization of plants and nutrients in ecosystems with adverse abiotic conditions, and ecosystem states develop with localized structures, such as gaps, labyrinths, stripes, and spots [55,56]. Once resource scarcity reaches a threshold, the ecosystem shifts toward a homogeneous state following the complete loss of plants [43]. In the early stages of vegetation fragmentation, patches are composed of large size, small patch numbers, high connectivity, and small total perimeter per unit area. Then an increase in environmental pressure causes continuous fragmentation of large patches into regular patch patterns made of large patch numbers, moderate size, and large total perimeter. Finally, still heavier environmental pressure drives the collapse of patches. Patches will disappear rapidly and a few small patches with small perimeters remain.

Slope position and aspect are considered among the most key abiotic factors influencing the spatial variability of soil properties via the pedogenic process on a large scope [57]. Slope position and aspect, therefore, lead to factors like heat, light, air, and water, which tend to affect soil properties. Temperature and precipitation along elevation gradients are the major factors affecting the carbon budget by controlling carbon inputs from plants

biomass and decomposition and this affects the stability of soil aggregates. Increased elevation improves precipitation thereby enhancing water availability and potentially aiding vegetation development. High moisture and low temperature decrease microbial activity and hence slow decomposition and mineralization rate of SOC. SOC was significantly higher on the shady slope than on the sunny slope in the study area. This could be attributed to the low mineralization of soil organic matter (SOM) due to the relatively low temperature, high humidity, and high moisture content on the shady slope. This is in line with [58]. The wetlands site under the KH patch also recorded a significantly high SOC content, and this is attributable to the elevated moisture content which reduces temperature and also tends to reduce microbial activity, and hence, there is a reduction in the rate of SOM mineralization rate. Under normal conditions, SOC distribution is affected by environmental conditions such as climate, vegetation, and soil texture [32]. The results also showed that soil moisture was high on the shady slope compared to the sunny slope. However, the soil temperature on the sunny slope was higher than on the shady slope. This likely is related to weak solar radiation on the shady slope and therefore less evapotranspiration. Changes in soil moisture have an impact on microbial activity [59]. Drought causes osmotic stress in soil microorganisms in dry pores; bacteria in saturated soils have better access to soil organic carbon and nutrients because of the well-connected soil pore network [60]. In field-moist soils, when the hydrologic connection is reduced and certain pores remain unconnected, this free exchange is reduced [61].

The results of this study equally showed that soil microbial biomass C and N significantly increased with increasing SOC and TN which is consistent with previous studies [62]. Variations in environmental conditions can influence C and N processes both via changes in abiotic actors and via changes in microbial community structure [63]. An induced switch in soil microbial community composition occurs by maximizing water and nutrient availability [64]. It can be seen from the results that the IL, KH, and BS patches under the low soil moisture sites recorded low levels of SOC, TN, $NO_3$-N, and $NH^+_4$-N among others. It is therefore not surprising that sites with low soil nutrients recorded patch configurations indicating they were unstable and undergoing degradation because inadequate nutrients and soil moisture put stress on vegetation resulting in its degradation.

## 5. Conclusions

To sum up, the study without a doubt indicates that topography and environmental conditions had significant effects on soil properties and spatial vegetation patch patterns and their configurations. RDA analysis showed that soil moisture is one of the most single factors that explained most of the variation in vegetation characteristics. Soil moisture is undoubtedly affected by elevation and slope aspects and should be taken into consideration in the selection and planning of vegetation restoration processes. The *Iris lactea* patch within the high soil moisture sites recorded the highest patch number per m$^2$ and total perimeter per m$^2$. Within the low soil moisture sites, *Kobresia humilis* patch had the highest patch number, and the *Iris lactea* patch recorded the highest fraction of vegetation cover and the largest total perimeter per m$^2$. The *Elymus nutans* patch under the sunny slope had a high patch number and total perimeter while the shady slope recorded a low patch number and small total perimeter. The *Iris lactea, Kobresia humilis,* and *Blysmus sinnocmpressus* patches under low soil moisture sites recorded patch configurations that show they are in unstable states and undergoing degradation. Therefore, urgent restoration attention must be given to those sites to prevent them from further degradation and its resultant effect of desertification. The *Kobresia humilis* patch under the low soil moisture areas recorded the highest values of Shannon–Wiener diversity index and Simpson's index of diversity probably due to the low plant coverage which allowed less competitive species to survive and grow. The highest value of soil organic carbon was recorded in the *Elymus nutans* patch located on a shady slope which could be attributed to the impact of elevation on soil-organic carbon. Temperature and precipitation along elevation gradients are the major factors affecting the carbon budget by controlling carbon inputs from plants' biomass

and decomposition. Increased elevation improves precipitation thereby enhancing water availability and potentially aiding vegetation development. Plant coverage on the shady slope was significantly higher than that of the sunny slope. This could be because of evapotranspiration within the sunny slope which is higher than that of the shady slope, and hence, species within the shady slope have adequate moisture for plant growth. The results would provide quantitative easy-to-use indicators for vegetation degradation and help in vegetation restoration projects. It would be essential to do more studies to fully comprehend the water balance in the study area on the connections between vegetation, soil properties, and topography to enhance the selection of suitable species for restoration programs.

**Author Contributions:** Conceptualization, T.A.A. and W.C.; methodology, T.A.A.; software, W.L.; validation, W.C., C.A.-A.W. and T.A.A.; formal analysis, T.A.A. and W.L.; investigation, T.A.A., S.W. and X.D.; resources, W.C.; data curation, C.A.-A.W. and W.L.; writing—original draft preparation, T.A.A.; writing—review and editing, W.C. and C.A.-A.W.; visualization, T.A.A. and W.L.; supervision, W.C.; project administration, W.C.; funding acquisition, W.C. All authors have read and agreed to the published version of the manuscript.

**Funding:** This research was supported by The National Natural Science Foundation of China (32060269). China Agricultural Research System of MOF and MARA (CARS-34).

**Institutional Review Board Statement:** Not applicable.

**Informed Consent Statement:** Not applicable.

**Data Availability Statement:** Data for this work are available upon request from the corresponding author.

**Acknowledgments:** We thank Wanting Liu and Wenhu Wang for their help in the laboratory analysis.

**Conflicts of Interest:** The authors declare no conflict of interest exist for this work.

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
