# Peer review of "Spatial Vegetation Patch Patterns and Their Relation to Environmental Factors in the Alpine Grasslands of the Qilian Mountains"

_sustainability, doi:10.3390/su14116738_

Round 1

Reviewer 1 Report

Overall the studies concept is good.  Justification for the study is  sound but trying to stretch your justification to restoration with out any context requires more explanation.  Some of your sentences are poorly structure and leaves the reader unclear as to the meaning. For example, line 41, you refer to "this area" without first defining what area you are talking about. 

Your methods are difficult to follow.  A diagram would be helpful. I'm not sure what you are calling replication.

You analysis is acceptable but not well articulated in your results section. I'm sure you have accurately explained the results in your discussion section.

Author Response

Responses to Reviewer 1 comments

Dear Editor,

The concerns and suggestions of the reviewers have been addressed to suit the requirement of your esteem journal. Below are the reviewers’ comments and authors response.

Comment 1

Justification for the study is sound but trying to link the justification to restoration without any context requires more explanation

Response

Thanks so much for the comment. Spatial vegetation patch patterns are indicators of ecosystem stability and health. Stable patches tend to have large size, small patch numbers, high connectivity, and small total perimeter per unit area. Detecting signs of regime changes are important to foresee and take measures to prevent the desertification of grasslands by developing recovery processes that enhance their sustainability. All these have been laid in the introduction, the results indicated that spatial vegetation patch configurations within the low soil moisture sites recorded patch configurations showing the are unstable and undergoing degradation. We equally extensively discussed this. It is therefore based on the discovery that spatial vegetation patches within the low soil moisture sites recorded patch configurations showing they are unstable and undergoing degradation that we suggested restoration process to enhance their sustainability.

Comment 2

Some of the sentences are poorly structured and leaves the readers unclear to the meaning. For example, in line 41. You refer to “this area” without first defining what area you are talking about.

Response

Thanks for the comment. It has been accordingly corrected and we read through and ensured that similar issues are corrected.

Comment 3

Your methods are difficult to follow. A diagram would be helpful. I’m not sure what you mean by replication.

Response

The diagram has been added. The replication talks about the number of repetitions done in the sampling. For instance, in this study, 10 random quadrats were laid in species patch per each selected site and repeated 3 times. Meaning if you take Kobresia humilis patch within the wetlands, first sampling point you lay 10 quadrats, second sampling point 10 quadrats and third point 10 quadrats.

Comment 4

Your analysis is acceptable but not well articulated in your results section. I’m you accurately explained the results in your discussion.

Response

Thanks so much for the suggestion. We read through and ensured that details of the results that were left out has been captured.

General Response

The revisions and corrections have been done and the content and format of the paper have been improved based on your suggestions.

We hope it meets your kind consideration.

                                                                                                Sincerely yours,

                                                                                                Abalori Theophilus Atio

Reviewer 2 Report

I enjoyed reading the manuscript. Hoever, there is text organizational issues in the text.

For example: Important values are described in line 169 but its table was given much earlier in L107.

"Patch Pattern" must be described first in the introduction section. Is this similar to spatial pattern? why patch? Sustainability journal has a broad reader group and descriptions are key for the readers to follow the text.

Table numbering issues: Table 3. 1. Major .. (is this Table 1?)

Table 2 was not discussed but only cited in the text.

Any relation with Table 3 and Figure 3 since both shows correlations? Are both necessary, why? Please justify.

Conclusions section: The realtion between soil moisture and precipitation / potential evapotranspiration patterns / field capacity / soil type etc must be highlighted in this section.
Readers can be cruious about hydrologic processes affecting vegetation dynamics like leaf area index and latitude etc. Currently most of the conclusion sentences are general / vague sentences.

Author Response

Responses to Reviewer 2 comments

Dear Editor,

The concerns and suggestions of the reviewers have been addressed to suit the requirement of your esteem journal. Below are the reviewers’ comments and authors response.

Comment 1

I enjoyed reading the manuscript. However, there are text organizational issues. For example: important values are described in line 169 but the table is given much earlier in line 107.

Response

It has been corrected and the description taken to line 107 where the table is located. We equally read through and ensured that similar issues are corrected.

Comment 2

“Patch pattern” must be first described in the introduction section. Is this similar to spatial patterns? Sustainability journal has a broad reader group and descriptions are key for the readers to follow the text.

Response

It has been corrected in the entire article. It is supposed to read “spatial vegetation patch patterns”.

Comment 3

Table numbering issues: table 3.1 major (is this table 1?)

Response

The numbering has been duly corrected. It is supposed to be Table 1.

Comment 4

Table 2 was not discussed but only cited in the text.

Response

Table 2 has been discussed as suggested.

Comment 5

Any relation with table 3 and figure 3 since both shows correlations? Are both necessary, why? Please justify.

Response

Thanks for the suggestion. Table 3 have been deleted. This is so because the RDA reveals the relative contributions of each variable to species composition data (See table 5).

Comment 6

Conclusions section: The relation between soil moisture and precipitation/potential evapotranspiration patterns/ field capacity/ soil type etc must be highlighted in this section. Readers can be curious about hydrologic processes affecting vegetation dynamics like leaf area index and latitude etc. Currently most of the conclusion sentences are general /vague.

Response

Thanks for the suggestion. The conclusion has been revised based on your suggestion.

General Response

The revisions and corrections have been done and the content and format of the paper have been improved based on your suggestions.

We hope it meets your kind consideration.

                                                                                                Sincerely yours,

                                                                                                Abalori Theophilus Atio

                                                                                                (abalorit@yahoo.com)

Round 2

Reviewer 1 Report

Revisions have improved the manuscript. 

Just a few specific edits.  Line 69  - either 2 sentences or change , to ;

Line 200 the word axes should follow after (DCA)

Not sure that lines 403-405 add to the discussion in this paragraph.

Author Response

Response to Reviewer 1 comments/suggestions

Comments:

Comment 1: Line 69 – either 2 sentences or change, to

Response: the sentence “Patchiness is regarded as a reflection of the state and functioning ability of ecosystems [12], patch sizes and types indicate major variations in soil attributes, plant biomass, and soil moisture [18,19].” Has been divided into 2 sentences as suggested. Thus, “Patchiness is regarded as a reflection of the state and functioning ability of ecosystems [12]. Patch sizes and types indicate major variations in soil attributes, plant biomass, and soil moisture [18,19].”

Comment 2: Line 200- the word axes should follow after (DCA)

Response: the word “axes” has been added immediately after “(DCA)” as suggested. Thus, “The first two Detrended correspondence analyses (DCA) axes explained 12.9% and 11.9% of the variation in species distribution of the selected grassland patches (Figure 2).”

Comment 3: Not sure that lines 403 – 405 add to the discussion in this paragraph

Response: Please, the sentences at lines 403 -405 have been deleted as suggested. Thus “Furthermore, species-specific characteristics may influence spatial pattern formation in a variety of ways. Large species, for example, have a strong influence on the establishment of patch size distributions [50]” highlighted yellow should be deleted. Try deleting the sentence with track changes on but wasn’t able so I coloured it.

Reviewer 2 Report

Thanks for the revised version 

Author Response

Thanks so much for your comments and suggestions.